# Diagnosis of the Severity of Fusarium Head Blight of Wheat Ears on the Basis of Image and Spectral Feature Fusion

**DOI:** 10.3390/s20102887

**Published:** 2020-05-20

**Authors:** Linsheng Huang, Taikun Li, Chuanlong Ding, Jinling Zhao, Dongyan Zhang, Guijun Yang

**Affiliations:** 1National Engineering Research Center for Agro-Ecological Big Data Analysis & Application, Anhui University, Hefei 230601, China; linsheng0808@ahu.edu.cn (L.H.); p18201092@stu.ahu.edu.cn (T.L.); p17301155@stu.ahu.edu.cn (C.D.); zhaojl@ahu.edu.cn (J.Z.); zhangdy@ahu.edu.cn (D.Z.); 2Beijing Engineering Research Center of Agricultural Internet of Things, Beijing 100097, China

**Keywords:** *Fusarium* head blight, imaging hyperspectral, spectral feature, texture feature, color feature

## Abstract

*Fusarium* head blight (FHB), one of the most prevalent and damaging infection diseases of wheat, affects quality and safety of associated food. In this study, to realize the early accurate monitoring of FHB, a diagnostic model of disease severity was proposed based on the fusion features of image and spectral features. First, the hyperspectral image of FHB infected in the range of the 400–1000 nm spectrum was collected, and the color parameters of wheat ear and spot region were segmented based on image features. Twelve sensitive bands were extracted using the successive projection algorithm, gray-scale co-occurrence matrix, and RGB color model. Four texture features were extracted from each feature band image as texture variables, and nine color feature variables were extracted from R, G, and B component images. Texture features with high correlation and color features were selected to participate in the final model building parameters via correlation analysis. Finally, the particle swarm optimization support vector machine (PSO-SVM) algorithm was used to build the model based on the diagnosis model of disease severity of FHB with different combinations of characteristic variables. The experimental results showed that the PSO-SVM model based on spectral and color feature fusion was optimal. Moreover, the accuracy of the training and prediction set was 95% and 92%, respectively. The method based on fusion features of image and spectral features can accurately and effectively diagnose the severity of FHB, thereby providing a technical basis for the timely and effective control of FHB and precise application of a pesticide.

## 1. Introduction

Wheat is the second largest food crop in China and one of the three largest food crops in the world, making it important both socially and economically. Stable and high yields of wheat have always been the focus of agricultural production in China, so timely and effective disease control is an important guarantee to achieve this goal [1]. *Fusarium* head blight (FHB; also known as wheat ear withered, rotten wheat head) is one of the main diseases of wheat, mainly caused by *Fusarium*
*graminearum* Schwabe and other *Fusarium* species [2,3]. After infection, the fungus can interfere with the normal physiological functions of wheat ears, causing changes in their morphology and internal physiological structure [4,5]. These changes cause serious effects on wheat. The occurrence of FHB can cause the large decline of wheat yield and quality reduction. Furthermore, *Fusarium* will produce secondary metabolites, particularly deoxynivalenol, which can cause human and animal poisoning and seriously endanger human and animal food safety and health [6,7]. Early detection of FHB can solve ecological environment problems caused by excessive use of chemical agents and achieve the purpose of timely control. Meanwhile, reducing the incidence of FHB and preventing the spread of the disease in a large area can effectively improve the wheat yield [8,9].

Frequently used methods for the detection of FHB include high-performance liquid chromatography, gas chromatography, molecular techniques, enzyme-linked immunoassay, and other chemical and biological detection methods. However, these methods have disadvantages such as long time-consuming process and large consumption of human and material resources, which cannot achieve rapid, non-destructive testing [10]. In recent years, the detection of image processing technology based on RGB images has provided new concepts and methods for plant diseases. Given that color imaging is inexpensive and easy to handle, it can be used to capture the color and texture information of objects. Color imaging has been used to detect and evaluate plant diseases. Cambaza et al. [11] found that color imaging can be implemented to detect *Fusasrium* based on the color differences between healthy and infected oat grains. Jr., D.G. Sena et al. [12] used digital images to distinguish maize plants affected by armyworm disease from healthy plants. Using visual analysis based on threshold segmentation to classify 720 images, the accuracy reached 94.72%. These results showed that plant diseases can be detected by image visual analysis. With the development of the deep learning network, researchers have proposed a large number of methods to detect plant diseases and classify disease levels. Qiu et al. [13] used a deep convolutional neural network (Mask RCNN) model using transfer learning to monitor FHB, and the mean average precision for the testing dataset was 0.921. Zhang et al. [14] proposed the IABC-K-PCNN method based on the fully convolutional network (FCN) segmentation of wheat ear to identify FHB. The method presented an accuracy of 0.925, indicating that it could effectively grade FHB in the field environment. However, these methods focused only on the surface of the image and did not consider changes in internal spectral information during plant infection.

The spectrum of diseased plant tissues can effectively reflect the changes in the chlorophyll content, water content, structure, and other information during plant infection [15]. Recently, hyperspectral imaging technology has been widely used in the direction of nondestructive testing and quality analysis of agricultural products, and its application in crop pest monitoring and identification has also been well developed [16,17,18]. Du et al. [19] used hyperspectral images to extract spectral feature band and employed the improved Gramsmite algorithm and genetic non-information variable algorithm, combined with linear discriminant analysis, stochastic forest, and support vector machine nearest node algorithm to establish a wheat deoxidized Fusarium alcohol content model for grade prediction with accuracy higher than 85%. Ding et al. [20] used interval combination optimization combined with successive projection algorithm to extract spectral features and constructed an FHB vomit toxin content prediction model. The correlation coefficient and mean square error of the best model were 0.921 and 0.375, respectively. Deng et al. [21] used hyperspectral images to fuse rice seed identification models established by spectral, texture, and morphological features to obtain satisfactory recognition accuracy, with a training set and test set accuracy of 99.22% and 96%, respectively. Lu et al. [22] used hyperspectral and image feature fusion to establish a lettuce disease pair model. The color and texture features of the sample images were extracted. The results showed that the model based on hyperspectral and image feature fusion performed well, and the accuracy of disease judgment was 92.23%.

On the basis of the above analyses, we collected wheat ear samples with different disease severity; further, we extracted the spectral features representing the components of the internal structure of the sample and the color and texture features reflecting the external properties [23]; and synthesized the spectral, texture, image, and other feature information of the hyperspectral image. We combined the particle swarm optimization–support vector machine (PSO-SVM) algorithm to establish the diagnosis model of FHB disease severity based on single feature and different feature fusion. Finally, we established an optimal diagnosis model of FHB disease severity through fusion of image and spectral features and the details discussed in Section 2.3.5.

## 2. Materials and Methods

### 2.1. Study Area and Data Collection

The experiments were launched in late April and early May 2019 at the experimental base of the Anhui Academy of Agricultural Sciences (117°14’ E, 31°53’ N). The experimental wheat (variety: Rotational 987) was sown in October 2018 and irrigated by conventional fertilization in all experimental plots. The *Fusarium graminearum* Sehw was inoculated in the flowering period of wheat (20 April 2019), and a spraying machine was used to spray it in the experimental plot (Figure 1c) to inoculate FHB. The details are shown in Figure 1. Two-hundred-and-sixty samples of diseased wheat ears were collected randomly during the wheat grouting period (1 May 2019). One-hundred-and-fifty samples were selected for the experiment after processing. To prevent the difference in spectral and image information from affecting the results, the sample with similar severity was selected, and only one-sided disease was studied.

### 2.2. Acquisition and Preprocessing of Hyperspectral Images

The hyperspectral image acquisition system consisted of an ImSpector V10E imaging spectrometer, a digital CCD camera, a zoom lens, a halogen lamp, a computer, and a sealed black cabinet (Figure 2). To reduce the interference of the light source and uneven illumination in the external environment, the whole acquisition process was carried out in the sealed black cabinet, and the halogen lamp was fixed at 45° on both sides of the sealed dark box. The height could be freely adjusted. When the hyperspectral images were collected, the ImSpector V10E imaging spectrometer was fixed above the black box cabinet, the lens was vertically facing downwards, and the wheat ear samples were tiled on the black cloth. To avoid image distortion, after multiple adjustments, the spectral channel was set to 256, and the image resolution was 696 × 696 pixels. The lens was determined to be 45 cm above the wheat ears, and the exposure time was 36 ms, ensuring that the image could be clearly imaged. All data processing was completed under the MATLAB platform.

During image acquisition, the standard whiteboard and full black calibration images were collected once to perform spectral correction on the original image. The correction formula is expressed as follows:(1)$$R=\frac{R_{s}-R_{w}}{R_{w}-R_{b}}$$

Among them, *R* is the corrected hyperspectral image, $R_{s}$ is the original spectral image, $R_{w}$ is the whiteboard image, and $R_{b}$ is the blackboard image.

### 2.3. Data Analysis Methodology

#### 2.3.1. Image Segmentation and Severity Calculation

To realize the early monitoring of FHB, the background of corrected hyperspectral images needs to be removed. The threshold-based image segmentation algorithm is a classical image segmentation method to segment the target object (wheat ear) with the background by finding the appropriate threshold. The HSV color space is a color space created based on the intuitive properties of colors. It consists of three components: hue (H), saturation (S), and value (V), which are relatively close to human visual properties, where hue (H) means different colors, saturation (S) refers to the depth of color, and value (V) represents the degree of light and shade of color. Compared with the RGB color space, the HSV color space can intuitively express the light and shade of the color, the hue, and the degree of value to facilitate the contrast between colors [24].

In this study, three wavelength images located at wavelengths of 680, 560, and 481 nm, which are closest to the three bands of red, green and blue were selected to synthesize RGB images. The wheat ear region in the sample image was segmented by the threshold method, and the segmented wheat ear region was used as the region of interest (ROI) to participate in the subsequent analysis. Thereafter, the hue (H) component in the HSV color space was used to set the automatic threshold to segment the spot area on the wheat ears.

The classification of disease severity was based on the classification index of occurrence degree stipulated in the technical specification for testing and reporting of FHB [GB/T15796-2011] and combined with the classification standard mentioned in previous studies [25,26]. The number of pixels in the area of disease spot extracted from a single wheat ear and the percentage corresponding to the number of pixels in the whole area of wheat ears served as the division standard (called the incidence rate of a single wheat ear). The severity of wheat infection with FHB was divided into five grades with X indicating the incidence, and the corresponding criteria for the five grades were: (1) 0 < X ≤ 10%, (2) 10% < X ≤ 20%, (3) 20% < X ≤ 30%, (4) 30% < X ≤40%, and (5) X > 40%.

#### 2.3.2. Spectral Feature Extraction

To reduce the influence of the noise of the external environment and the instrument itself on the spectral data, the original spectral data must be preprocessed. In this paper, orthogonal signal correction was used to preprocess the spectral matrix. The mathematical orthogonal method was used to filter out the information in the original spectral matrix that was not related to the quality to be tested [27].

Successive projection algorithm (SPA) is a forward variable selection algorithm that minimizes the collinearity of vector space. It employs projection analysis among vectors to eliminate the redundancy in the original spectral matrix, minimize the collinearity between variables, and determine the combination of variables with minimal redundancy information [28]. *k* (0) is the initial wavelength, and N is the number of wavelength variables extracted. The algorithm starts at one wavelength and calculates its projection on the remaining other wavelengths. The wavelength variable with the largest projection vector is added to the wavelength combination, and cycle n times ends [29]. The algorithmic steps for SPA are as follows [30]:

Step 1: Before the first iteration (*n* = 1) begins, record any vector in the spectrum matrix $X_{I\times J}$ (*I* is the number of samples, and *J* is the number of wavelengths) as $x_{\left(k0\right)}$; $x_{\left(k0\right)}$ is the initial wavelength iteration vector.

Step2: Record the unselected wavelength set as follows:(2)S={j,1≤j≤J,j∉{k(0), ⋯,k(n−1)}}

Step 3: Calculate the projection of $x_{j}$ to the vectors in set *S*, that is, calculate the projection of the spectral data of the initial wavelength variable orthogonal to the spectral data of other wavelength variables as follows:(3)$$Px_{j}=x_{j}-\left(x_{j}{}^{T}x_{k\left(n-1\right)}\right)x_{k\left(n-1\right)}{\left(x_{k\left(n-1\right)}{}^{T}x_{k\left(n-1\right)}\right)}^{-1}$$

Step 4: Record the largest projection value in *N*−1 projection as *k*(*n*):(4)$$k\left(n\right)=arg(max(\|Px_{j}\|),j\in S)$$

Step 5: Use the maximum projection as the initial value of the next iteration as follows:(5)$$x_{j}=Px_{j},j\in S$$

Step 6: *n* = *n* + 1 if *n* < *N* return Step 2.

Finally, the combination of variables is {*k*(*n*); *n* = 0, …, *n*−1}. The optimal combination of wavelength variables is selected based on the magnitude of the value of RMSE for each *k* (0) and *n* determined.

#### 2.3.3. Texture Feature Extraction

Texture feature is a kind of visual feature of homogeneous phenomenon in a reactive image, which is a common and difficult-to-describe feature in the image. It embodies the surface structure organization arrangement attributed to slow change or periodic change on the surface of an object, and it is commonly used for image classification and scene recognition. It has a certain effect on the early recognition of crop diseases and the improvement of their accuracy [31].

The gray level co-occurrence matrix (GLCM) was proposed by Haralick et al. [32] in the early 1970s. It is a broad texture analysis method based on the assumption that the spatial distribution relationship between pixels in the image contains image texture information. The method of extracting texture features based on GLCM is a classical statistical analysis technique and a currently recognized texture analysis tool [33]. In the present study, the GLCM algorithm was used to extract the texture features of the image to explain the differences between different disease severity wheat. Haralick et al. [32] extracted 14 characteristic components based on the gray-scale symbiosis matrix: energy, entropy, contrast, uniformity, correlation, variance, sum average, sum variance, sum entropy, difference variance, difference average, difference entropy, correlation information measure, and maximum correlation coefficient. However, Balaldi [34] reported that not all of the 14 texture feature components can be applied to the classification of images; contrast, energy, entropy, and correlation are easy to calculate and yield the best effects. In the present study, these four feature components were selected as texture features for subsequent modeling.

The principle of gray co-occurrence matrix is based on the pixel points at the point (*k*, *l*) of gray scale i in the image (size *m* × *n*). It describes the probability P(i,j,d,θ) [35] of the simultaneous occurrence of pixel points at the point (*m*, *n*) of gray level j at the point (*k*, *l*) with a distance of *d* orientation *θ* [35]. *d* is the relative distance expressed by the number of pixels (if *d* = 1, denotes neighboring pixels); *θ* usually considers four directions of horizontal, diagonal, vertical, and diagonal (0, 45°, 90°, and 135°, respectively) [36]. The mathematical formula is
(6)P(i,j,d,θ)={(k,l),(m,n)∈(M.N)|f(m,n)=i,f(m,n)=j}
where *I*, *j* = 0, 1, 2, …, *L*−1; *L* is the gray scale series of the image; *m* = 1, 2, 3, …, *M*; and *n* = 1, 2, 3, …, *N*.

(1) Contrast (CON): It directly reflects the brightness contrast of a pixel value and its domain pixel value, and it reflects the clarity of the image and the depth of the texture. The deeper the texture, the bigger the contrast, the clearer the effect; conversely, if the contrast value is small, then the grooves are shallow and the effect is blurred. It is expressed as follows:(7)$$\mathrm{CON}=\overset{k-1}{\underset{n=0}{\sum }}n^{2}\left\{\underset{\left|i-j\right|=n}{\sum }G\left(i,j\right)\right\}$$

(2) Energy (ASM): It is the sum of the squares of the elements of the gray-scale symbiosis matrix, reflecting the uniformity of the gray distribution of the image and the fineness of the texture. Small energy means a fine texture, whereas large energy indicates a rough texture. It is expressed as follows:(8)$$\mathrm{ASM}=\overset{k}{\underset{i=1}{\sum }}\overset{k}{\underset{j=1}{\sum }}{\left(G\left(i,j\right)\right)}^{2}$$

(3) Entropy (ENT): It is a random measure of the amount of information that an image has. The larger the entropy, the more complex the image is; conversely, the smaller the entropy, the simpler the image is. It is expressed as
(9)$$\mathrm{ENT}=-\overset{k}{\underset{i=1}{\sum }}\overset{k}{\underset{j=1}{\sum }}G\left(i,j\right)logG\left(i,j\right)$$

(4) Correlation (COR): It is a measure of spatial gray-level symbiosis matrix elements in the row or column direction of similarity, so the correlation value size reflects the image local gray correlation. The larger the phase value, the greater the correlation; conversely, the smaller the value, the smaller the correlation.
(10)$$\mathrm{COR}=\overset{k}{\underset{i=1}{\sum }}\overset{k}{\underset{j=1}{\sum }}\frac{\left(ij\right)G\left(i,j\right)-u_{i}u_{j}}{s_{i}s_{j}}$$
(11)$$u_{i}=\overset{k}{\underset{i=1}{\sum }}\overset{k}{\underset{j=1}{\sum }}i\cdot G\left(i,j\right)$$
(12)$$u_{j}=\overset{k}{\underset{i=1}{\sum }}\overset{k}{\underset{j=1}{\sum }}j\cdot G\left(i,j\right)$$
(13)$$s_{i}^{2}=\overset{k}{\underset{i=1}{\sum }}\overset{k}{\underset{j=1}{\sum }}G\left(i,j\right){\left(i-u_{i}\right)}^{2}$$
(14)$$s_{j}^{2}=\overset{k}{\underset{i=1}{\sum }}\overset{k}{\underset{j=1}{\sum }}G\left(i,j\right){\left(j-u_{j}\right)}^{2}$$

The texture feature extraction of feature band images was based on the determination of sensitive bands in Section 2.3.2. instead of the texture feature extraction of all 256 band images with high spectrum.

#### 2.3.4. Color Feature Extraction

Color features are the most widely used visual features in image retrieval. Color is often related to the objects or scenes contained in the image; meanwhile, the color features are less dependent on the size, direction, and angle of view of the image itself and have high robustness. In the early onset of FHB, wheat ears will produce light brown markings that gradually extend to the whole ears; eventually, the ears wither and become yellow. Therefore, the color difference can be used to distinguish the severity of FHB. In this paper, we used the wavelengths (680, 560, and 481 nm) of three images in the hyperspectral image to synthesize the RGB image as the target image and extract the color features. We calculated the color features of the three component images of the target images *R*, *G* and *B* (*R*, *G*, and *B* components represent the red, green, and blue components in the RGB color space, respectively). The color moment of the image is used to represent the color characteristic: first moment ($u_{i}$), second moment ($\sigma_{i}$), and third moment ($s_{i}$), as follows [37]:(15)$$u_{i}=\frac{1}{N}\overset{N}{\underset{j=1}{\sum }}P_{i,j}$$
(16)$$\sigma_{i}=(\frac{1}{N}\overset{N}{\underset{j=1}{\sum }}{\left(P_{i,j}-u_{i}\right)}^{2}{)}^{\frac{1}{2}}$$
(17)$$s_{i}=(\frac{1}{N}\overset{N}{\underset{j=1}{\sum }}{\left(p_{i,j}-u_{i}\right)}^{3}{)}^{\frac{1}{3}}$$
$p_{i,j}$ indicates the probability of the pixel with a gray level *j* in the *i-th* color channel component in the color image, and *N* indicates the number of pixels in the image. Therefore, for the three color components *R*, *G*, and *B* of the RGB image, each color component extracts three feature variables to form a nine-dimensional histogram vector. The color feature of the image is expressed as follows [38]:(18)$$F_{color}=\left[u_{R},\sigma_{R},s_{R},u_{G},\sigma_{G},s_{G},u_{B},\sigma_{B},s_{B}\right]$$

#### 2.3.5. PSO-SVM Modeling

Particle swarm optimization (PSO) was first proposed by Eberhart and Kennedy [39]. This global optimization algorithm was derived from the swarm movement behavior of birds and fish swarms. The basic idea of PSO algorithm is to find the optimal solution through cooperation and information sharing among individuals in the swarm. Its advantages include simple implementation and minimal parameter adjustment.

Support vector machine (SVM) is based on statistical learning theory and minimization of institutional risk, it has the advantages of high accuracy and strong learning ability in solving small samples and nonlinear and high-dimensional pattern recognition [40]. It uses the kernel function to map the sample data to a high-dimensional feature space and builds an optimal linear relationship model for solving. SVM parameter optimization is currently the most commonly used grid search method to obtain the optimal parameters, but this method has a large workload and low efficiency [41] and has a high dependence on the selection of parameters; the selection of kernel functions and the setting of penalty factors have extremely high requirements. The PSO algorithm is an evolutionary algorithm. It starts from a random solution, finds the optimal solution through iteration, and then evaluates the quality of the solution through fitness. However, it is simpler than the rules of genetic algorithms. It does not have the “Crossover” of genetic algorithms and “Mutation” operations to find the global optimum by tracking the currently searched optimal value. This algorithm has the advantages of easy implementation, high accuracy, and fast convergence speed. It has attracted widespread attention in the academic community and is superior in solving practical problems. This paper selected the radial basis function (RBF) as the kernel function, which can realize non-linear mapping, and the RBF kernel has less numerical difficulties [42]. The PSO was used to optimize the SVM. Parameters c and g were set to improve the stability and generalization ability of the model, where c is the penalty coefficient, that is, the tolerance of the error. The larger c is, the easier it is to overfit, and the smaller c is, the easier it is to be undersimulated; g is a parameter that comes with the function after selecting RBF as the kernel that determines the distribution of the data after it is mapped to the new feature space, which indirectly affects the speed of training and prediction. The algorithm steps for establishing a wheat ear disease severity diagnosis model using PSO-optimized SVM are as follows:(1)Initialize the particle population, including random positions and velocities.(2)Find the optimal solution through iteration. In each iteration, the particle updates itself by tracking two “extreme values” (pbest, gbest).(3)After finding these two optimal values, the particles follow their own speed and position.(4)Determine whether the maximum number of iterations initially set was met, and the optimal penalty factor and kernel parameters were obtained when the conditions were met.(5)Classification of wheat scab mildew severity using parameter optimized SVM model.

## 3. Results

### 3.1. Image Segmentation

Figure 3a presents the wheat RGB image of infected FHB, and Figure 3b is the gray histogram of this image. The appropriate gray value in the histogram was selected as the global threshold. The image was segmented by the threshold, the wheat ear region was extracted, and the binary image was obtained. Finally, morphology learning processing was carried out to remove the burr and noise, and the binary map of the wheat ear image with the separation background was obtained (Figure 3c). Using this binary image to mask the RGB image to remove background noise, the obtained RGB image contained only the region of interest (Figure 3d). The above segmented wheat ear area image was further transformed into HSV color space, and the H component in the HSV color space was used to set an automatic threshold to segment the diseased spot area. Figure 4a illustrates the H component map, in which the grayish white portion is the non-spot area, and the grayish black part is the diseased spot area. The segmentation results are shown in Figure 4b. The H component in the HSV color space could be used to segment the diseased spot area well.

### 3.2. Spectral Analysis of Wheat Ears with Different Severity

The original mean spectrum of wheat ear samples with different disease severity and the average spectrum after the correction of orthogonal signal are shown in Figure 5. A comparison of the original spectrum with the spectrum after pretreatment revealed that the classification boundary between the spectral information of different disease severity samples after pretreatment was clearer than that without pretreatment, and the effect was obvious. Figure 4 shows that the average spectral curve of the samples with different disease severity had the same overall trend, but the reflectivity showed an increasing trend with the increase in disease severity. In the range of 520–680 nm, the slope changes of the samples with different disease severity were different. At 670 nm, the red edge position began to appear with the increase in disease severity, which may be caused by the change in physicochemical parameters of wheat ears due to the occurrence of FHB.

### 3.3. Feature Variable Extraction

#### 3.3.1. Feature Wavelength Extraction

In this paper, the characteristic wavelengths of the sample spectral data of the training set were screened by the SPA algorithm (Figure 6a). The RMSE was 0.34933 when the number of selected variables was 12. Thereafter, the RMSE became stable. Therefore, 12 characteristic wavelength variables (442, 491, 552, 675, 685, 693, 698, 706, 757, 767, 924, and 935 nm) were selected for sample spectral data via SPA. The results are shown in Figure 6b.

#### 3.3.2. Texture Feature Extraction

Through GLCM, the contrast, energy, entropy, and correlation of the images corresponding to 12 feature bands were calculated, and 48 texture features were obtained. For each sample, the average texture features of the contrast, energy, entropy, and correlation of the images corresponding to 12 feature bands were calculated. Figure 7 shows the 12 feature band images.

#### 3.3.3. Color Feature Extraction

The three components of each sample image (R, G, and B) exhibited nine color moment characteristics: R component first moment, R component second moment, R component third moment, G component first moment, G component second moment, G component third moment, B component first moment, B component second moment, and B component third moment. Figure 8 shows the three-component result plots of wheat ear sample images R, G, and B.

### 3.4. Variable Screening and Model Building

#### 3.4.1. Selection of Characteristic Variables

The results of correlation analysis of texture and color features of samples with different disease severity are shown in Table 1. Table 1 shows that the correlation exhibited a strong relationship with the severity of FHB, and the correlation coefficient was −0.6969. Contrast, energy, and entropy showed a low correlation with the severity of the wheat condition, with correlation coefficients of 0.1716, 0.0755, and 0.2588, respectively. Therefore, the correlation was selected as the final texture feature. Twelve variables per sample were used as the texture features for the analysis of wheat condition severity. Correlation as a texture feature mainly reflected the local correlation of wheat with different disease severity.

Table 1 shows that the six color characteristics of R-component first-order moment, R-component second-order moment, R-component third-order moment, G-component first-order moment, G-component second-order moment, and G-component third-order moment were highly correlated with the severity of wheat condition, with correlation coefficients of 0.8480, 0.5428, 0.7723, 0.8097, 0.6028, and 0.6870, respectively. By contrast, the first-order moment of B component, the second-order moment of B component, and the third-order moment of B component were poorly correlated with the severity of wheat condition, and the correlation coefficients were 0.4542, −0.2513, and 0.2314, respectively. Therefore, this paper chose six color features with high correlation to analyze the severity of FHB. The first-order moment denotes the mean, the second-order moment denotes the variance, and the third-order moment denotes the slope.

#### 3.4.2. Modelling

A total of 100 and 50 data samples were used as training and validation sets, respectively. The PSO-SVM algorithm was used to establish the diagnostic models of different severity of FHB based on spectral features, texture features, and color features. Furthermore, the different feature values were standardized for fusion, and the diagnostic models based on spectral features plus color features, spectral features plus texture features, and spectral features plus color features plus texture features were established. The model prediction results and the optimized hyperparameter combinations are shown in Table 2 and Figure 9.

As shown in Table 2, the prediction results of constructing the PSO-SVM model based on single characteristic variables were obviously different. The spectral characteristic model obtained comprehensive and good results with accuracy of training and validation sets of 85% and 84%, respectively. These values were slightly better than the color characteristic model with training and validation set accuracies of 86% and 82%, respectively. Both models could be used to diagnose the severity of FHB. Thus, the internal spectral information and the external color information strongly contributed to the diagnosis of wheat disease severity with the increase in FHB. By contrast, the prediction results of the texture feature model were poor, and the accuracy of the training set and the validation set was 75% and 68%, respectively. These values indicated that the external texture features did not show a significant gradient difference with the increase in wheat condition.

An analysis of the prediction results of the PSO-SVM model after fusion of different eigenvalues revealed that the model results of spectral and color feature fusion were optimal and showed strong prediction ability. The accuracy of the training and validation sets was 95% and 92%, respectively. These values were better than spectra plus color plus texture feature model, and the accuracy of the training and validation sets was 85% and 82%, respectively. However, the prediction results of the spectral plus texture feature model were relatively poor, with a training and validation set accuracy of 82% and 78%, respectively. Consistent with the prediction results of the model based on a single feature, the results showed that the texture features extracted in this paper had a slight contribution to the diagnosis of wheat disease severity relative to the color features and spectral features. This analysis demonstrated that the wheat condition severity prediction model constructed by the fusion of spectral and color features and combined with the PSO-SVM algorithm showed optimal results. The prediction ability of this model was obviously better than that established by using single spectral or color features.

## 4. Discussion

FHB is common throughout the world and is mainly distributed in humid and semi-humid regions, especially in humid and rainy temperate areas. In recent years, many plant diseases occurred frequently during spring because of the increasing temperature and rainfall in China [43]. Thus, realizing early identification and monitoring of disease and pest infestations is of great significance. In previous studies, researchers had focused on the use of multitemporal and large-scale remote sensing data to analyze the impact of climate changes on the occurrence of plant diseases [44,45]. However, large-scale prediction methods based on climate and satellite remote sensing data are no longer applicable to the current precision agricultural-scale disease surveillance and estimation. Hyperspectral analyses can provide detailed information and have been proven to be considerable help in estimating plant diseases [10,46].

Most previous studies focused on a single variable of the image information or spectrum to study FHB. Zhang et al. [7] proposed a FHB classification index targeting FHB-infected ears using hyperspectral microscopy images and got the final identification accuracy of 89.80%. However, this method requires more sophisticated equipment. Zhang et al. [14] using Neural Network to identify the severity of wheat FHB, yielded a better classification (92.5%). This method has to extract a large amount of image feature information. This paper presented a method of using hyperspectral image by combining spectrum and image information to diagnose the disease severity of FHB and the classification accuracy reached 92%. Compared with the previous methods that use a single spectral feature or image feature information to identify, classify, and monitor crop diseases and insect pests, this study used hyperspectral image to diagnose the severity of FHB by combining the spectral information that represent the internal components of the sample with the image information that represent the external attributes. The results showed that with the increase of the severity of FHB, the internal spectral information and external color information had a greater contribution to the diagnosis of wheat disease severity, whereas the external texture feature information had a smaller contribution to the diagnosis of wheat disease severity. With the increase in the disease severity of FHB, the changes in water content, pigment, and cell structure in the inner cells alter the spectral response of wheat plants. Moreover, with the increase in disease severity, the area of dry and white ears formed by wheat ears becomes increasingly large, which is consistent with the research results. In this study, different features were fused, and the model that is based on the combination of spectral and color features was the best.

In this study, hyperspectral imaging technology was used to diagnose the severity of FHB. The severity of FHB was diagnosed by extracting the spectral information and the image information of external attributes. We opt to use the PSO to optimize the SVM model. The grid search method is the traditional method of SVM parameter optimization. However, this method has a large workload and low efficiency. PSO has the advantages of simple implementation, high accuracy, fast convergence, and minimal parameter adjustment. Given that many factors affect the spectral reflectance of crops, and different diseases will present different symptoms, the use of hyperspectral imaging technology to achieve rapid and accurate identification and classification of diseases and pests still needs further research.

## 5. Conclusions

FHB, as a typical disease of wheat, has a huge impact on grain harvest. In this study, hyperspectral images and combined spectral and image information were adopted to develop a novel method for the diagnosis of FHB. Using the PSO-SVM algorithm, the disease severity diagnosis model of FHB was established based on the fusion of spectral features, texture features, color features, and different features. For the collected spectral data, 12 characteristic wavelength variables were extracted by the SPA algorithm. For each sample, 12 texture feature variables were extracted by combining gray level symbiosis matrix and correlation analysis. On the basis of the RGB color model, color moment and correlation analysis were conducted to extract R, G, and B three-component maps for six color feature variables. Combined with the PSO-SVM algorithm, the model of FHB condition severity diagnosis with single feature and different feature fusion was constructed. The PSO-SVM model based on spectral and color feature fusion showed optimal results. The accuracy of the training and validation set was 95% and 92%, respectively; these values were 8% higher than that of the PSO-SVM model based on a single spectral feature. The results showed that use of hyperspectral images, fusion of spectral and color features, and combination with the PSO-SVM algorithm can effectively achieve the diagnosis of FHB disease severity and provide a new line of thought for the study of FHB.

## Figures and Tables

**Figure 1 sensors-20-02887-f001:**
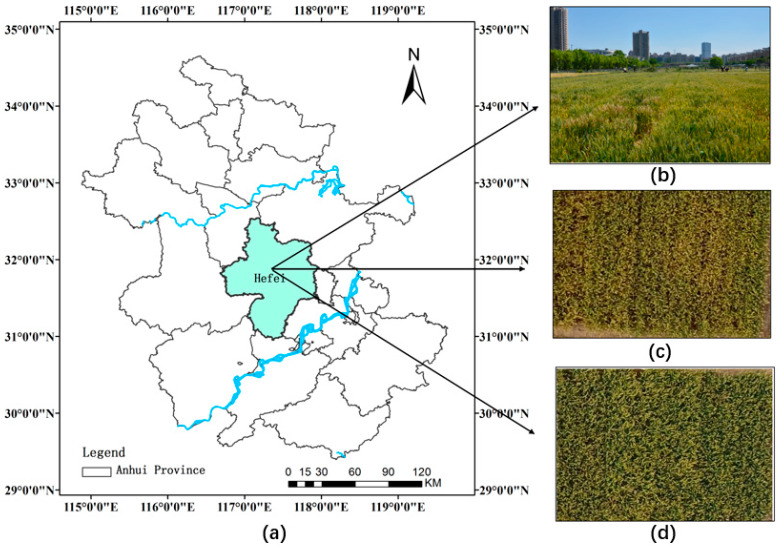
Experimental field. (**a**) Experimental site; (**b**) fieldwork situation; (**c**) vaccinated experimental area; (**d**) unvaccinated experimental area.

**Figure 2 sensors-20-02887-f002:**
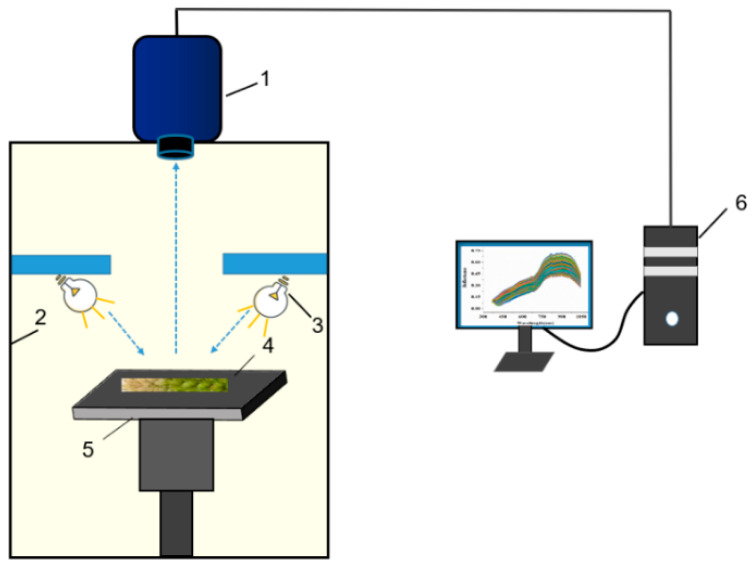
Composition of the hyperspectral system. (1) Imaging spectrometer; (2) Black cabinet; (3) Halogen lamp; (4) Samples; (5) Plain platform; (6) Computer.

**Figure 3 sensors-20-02887-f003:**
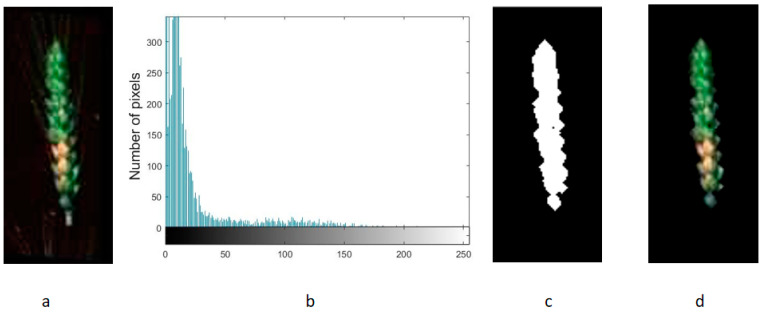
Segmentation of wheat ear. (**a**) The wheat RGB image of infected FHB; (**b**) The gray histogram of the image; (**c**) The binary map of the wheat ear image with the separation background; (**d**) The wheat RGB image contained only the region of interest.

**Figure 4 sensors-20-02887-f004:**
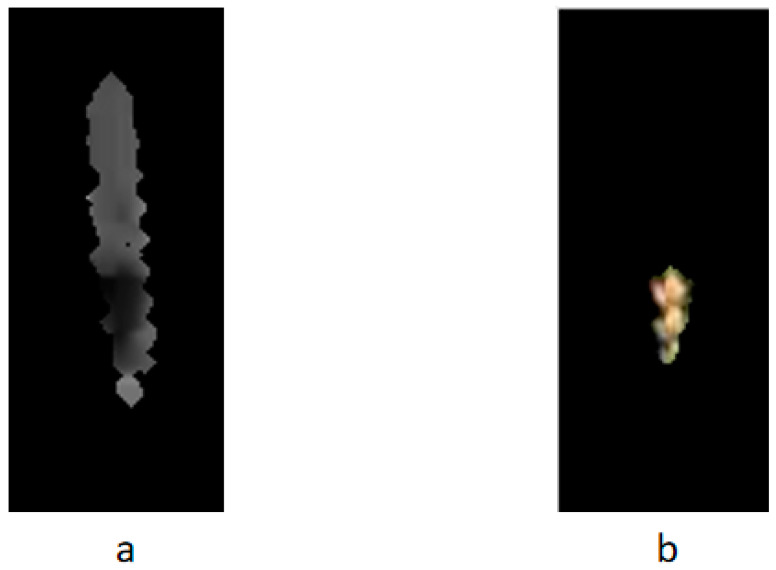
Segmentation of diseased spots. (**a**) The H component map of HSV color space; (**b**) The diseased spot area after segmentation.

**Figure 5 sensors-20-02887-f005:**
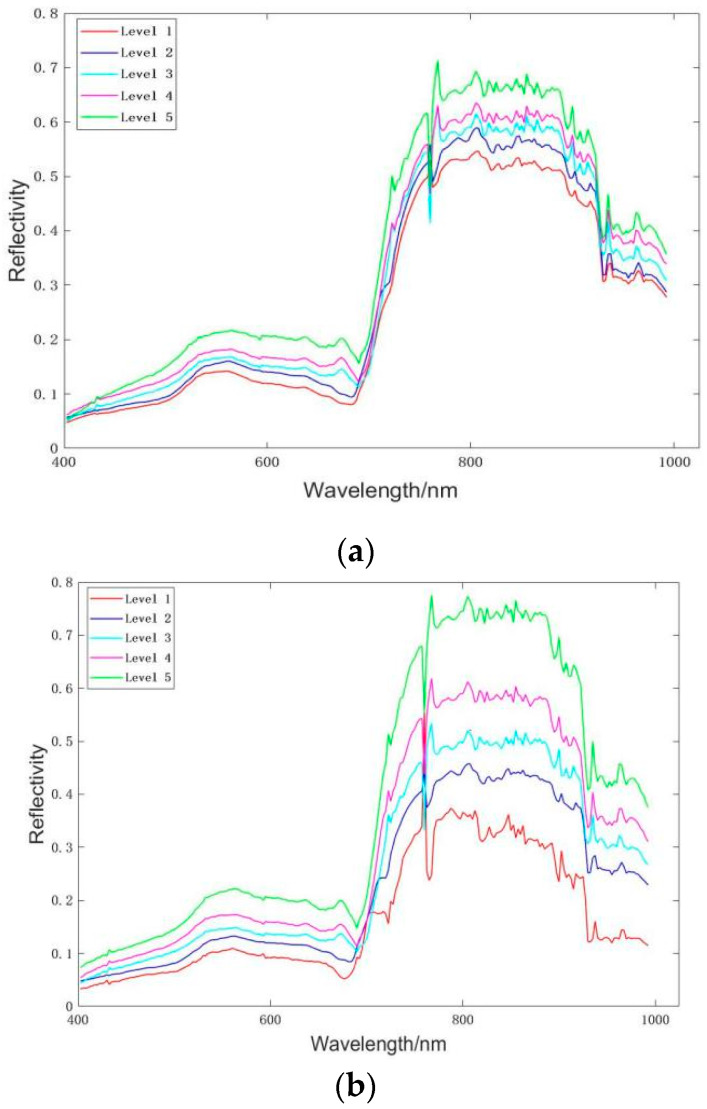
Spectral information. (**a**) Original spectral curve; (**b**) Spectral curve after pretreatment.

**Figure 6 sensors-20-02887-f006:**
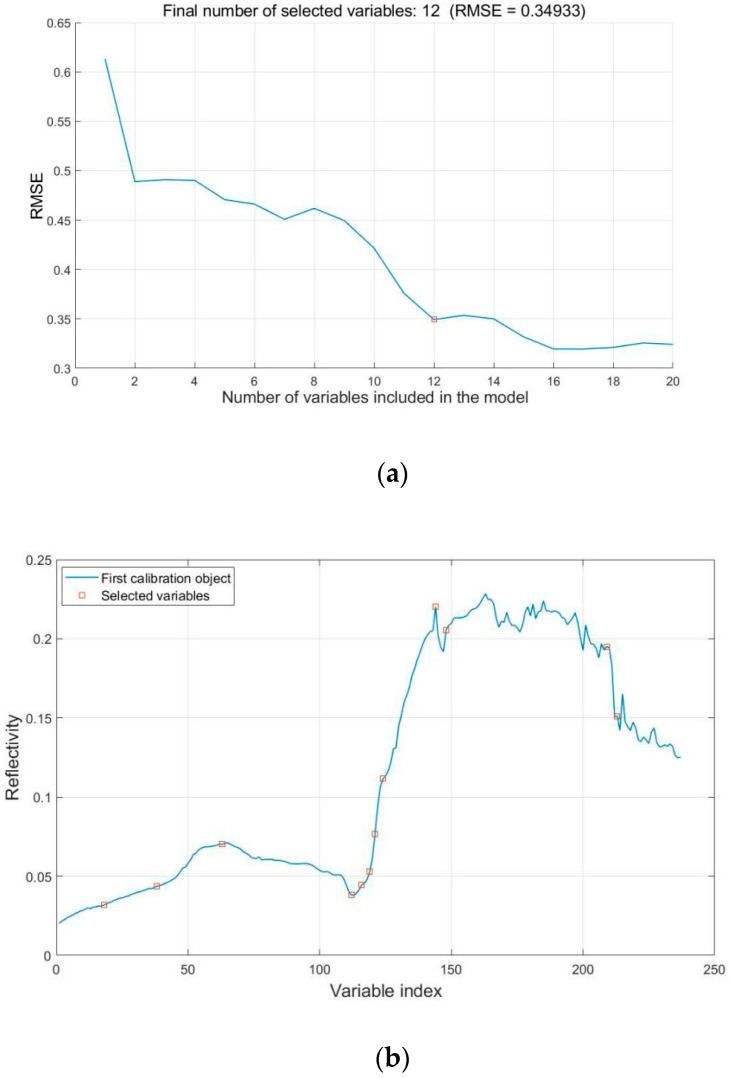
Feature wavelength screening results of the SPA algorithm. (**a**) The characteristic wavelengths of the sample spectral data of the training set; (**b**)The specific distribution of characteristic band.

**Figure 7 sensors-20-02887-f007:**
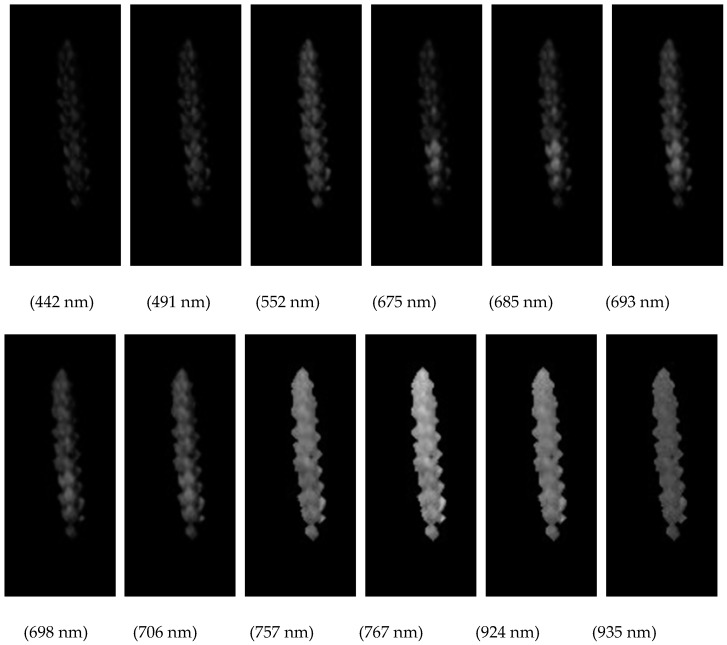
Twelve feature band images.

**Figure 8 sensors-20-02887-f008:**
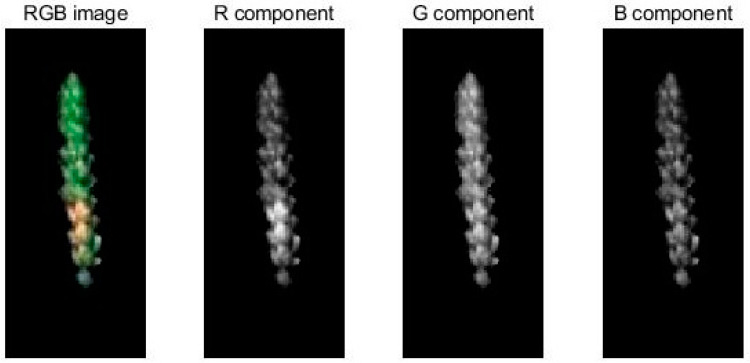
Results of three components of wheat ear samples: R, G, and B.

**Figure 9 sensors-20-02887-f009:**
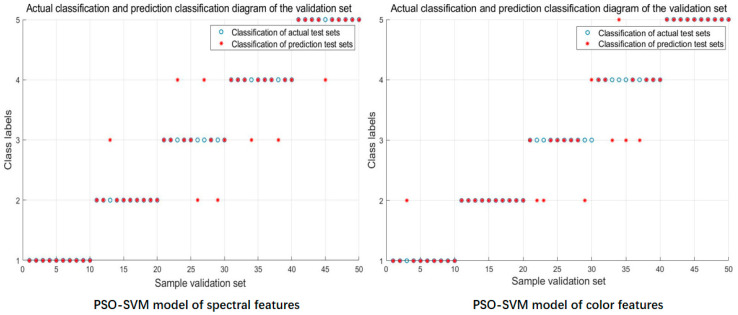
PSO_SVM model prediction results with different feature information.

**Table 1 sensors-20-02887-t001:** Results of correlation analysis of texture and color characteristics of samples with different severity of FHB.

Texture and Color Features	Correlation Coefficient
Contrast	0.1716
Energy	0.0755
Entropy	0.2588
Correlation	−0.6969
R-component first-order moment	0.8480
R-component second-order moment	0.5428
R-component third-order moment	0.7723
G-component first-order moment	0.8097
G-component second-order moment	0.6028
G-component third-order moment t	0.6870
B-component first-order moment	0.4542
B-component second-order moment	−0.2513
B-component third-order moment	0.2314

**Table 2 sensors-20-02887-t002:** PSO_SVM model prediction results for different feature information.

Feature Information	Model Set Accuracy	Validation Set Accuracy	c, g
Spectral features	85%	84%	20.2973, 14.6059
Color features	86%	82%	7.7751, 7.4498
Texture features	75%	68%	40.3025, 2.1227
Spectral + Color features	95%	92%	38.0265, 0.8672
Spectral + Texture features	82%	78%	3.9114, 3.1051
Spectral + Color + Texture features	85%	82%	99.3662, 0.0136

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
