# Peer review of "Diagnosis of the Severity of Fusarium Head Blight of Wheat Ears on the Basis of Image and Spectral Feature Fusion"

_sensors, 2020, doi:10.3390/s20102887_

Round 1

Reviewer 1 Report

The manuscript contains very important and interesting information and can be published after major revision.  

Names of species and genus should always be italicized (e.g. lines 38, 39, 42, etc., also in References).

Lines 48-52: Authors should also mention molecular methods.

Lines 57, 441: There is an error in the name. The name of this author is Sena Jr., D.G.

Line 100: What species of Fusarium was used to inoculate?

Subsection 2.1 Study Area and Data Collection should be described in more detail. What area was inoculated? How were the samples collected? Randomly? How many samples were collected?

Line 101: Why the authors mention bacteria? The genus Fusarium belongs to the fungi. This sentence is not understood.

Figure 1. Is there a typo? Should it be "Experimental field"?

Figure 1 is not cited in the text of subsection 2.1.

Line 140: It should be clearly indicated why these wavelengths were selected.

Line 185: It seems to me that quoting this article [31] is not correct. This article is not about crop diseases.

Lines 186, 192: "et al [32]." should be changed to "et al. [32]" (dot in the wrong place)

Line 193: It has been written, "variance, mean, variance". Should it be "variance, mean variance"? (without a comma)

Line 305: "in table 1" should be changed to "in Table 1"

Subsection 3.4.1. Selection of characteristic variables: Authors should indicate which correlations are significant. At what significance level.

I think that the section Discussion should be supplemented. The obtained results should be compared with the literature data. What were the results of the identification of Fusarium head blight of wheat and classification of Fusarium-infected and healthy wheat ears or wheat kernels based on features from hyperspectral images reported by other authors? Did the applied method provide higher accuracies compared to other methods?

Author Response

1. Names of species and genus should always be italicized (e.g. lines 38, 39, 42, etc., also in References).

    Revised

2.Authors should also mention molecular methods.

    Revised in Line 49

3. There is an error in the nameThe name of this author is Sena Jr., D.G

    Revised in Line 57

4. What species of Fusarium was used to inoculate?

    The Fusarium graminearum Sehw  revised in Line 100

5. Study Area and Data Collection should be described in more detail. What area was inoculated? How were the samples collected? Randomly? How many samples were collected?

   Revised in Section 2.1

6. Why the authors mention bacteria? The genus Fusarium belongs to the fungi. This sentence is not understood.

   Revised in Line 101

7.Is there a typo? Should it be "Experimental field"?

   field Revised 

8.Figure 1 is not cited in the text of subsection 2.1.

   Revised in Line 102

9.It should be clearly indicated why these wavelengths were selected.

   Revised in Line 143

10.It seems to me that quoting this article [31] is not correct. This article is not about crop diseases.

    Revised in Line 490

11.Lines 186, 192: "et al [32]." should be changed to "et al. [32]" (dot in the wrong place)

    Revised

12.It has been written, "variance, mean, variance". Should it be"variance, mean variance"? (without a comma)

   Revised in Line 196~198

13. in table 1" should be changed to "in Table 1" 

    Revised

14.Selection of characteristic variables: Authors should indicate which correlations are significant. At what significance level.

    Line 308-312 / Line 315-319 shows the which correlations are significant. This section mainly selects the large correlation coefficient as the feature.

15. I think that the section...

      Revised in 376-383

Reviewer 2 Report

This is a potentially interesting subject and a topic that seems suitable for the selected outlet. However, the main novelty of paper is supposed to be PSO-SVM for evaluating disease severity of FHB using hyperspectral images. But in fact, the authors describe the well-developed methods that previously was used. So, the novelty of the paper is not fully exploited and authors should considerably improve and strengthen the novelty of the paper.

LL.91-93
There’s poor review of the proposed algorithm. Consequently, there’s no clear research gap identified that this study is filling.
You need to clarify the reason why use used PSO-SVM.

LL. 186-188
Did you copy these sentences from Li et al (2018)?

LL. 226-228
Did you copy from Lu et al (2019)?

2.3.5. Modeling
Why did you choose RBF kernel?
Which software did you use?
How did you optimize the hyperparameters?

3. Results
The combinations of SVM-hyperparameters must be presented for each method.

Table 1
You showed the results of correlation analysis. Did you evaluate any reciprocal cooperation?

4. Discussion
A deeper discussion of the results in relation with similar works in literature in needed.

Author Response

1.There’s poor review of the proposed algorithm. Consequently, there’s no clear research gap identified that this study is filling.
You need to clarify the reason why use used PSO-SVM.

  Revised in line 94

2. Did you copy these sentences from Li et al (2018)?

   No.

3.Did you copy from Lu et al (2019)?

   No.

4. 2.3.5. Modeling
Why did you choose RBF kernel?
Which software did you use?
How did you optimize the hyperparameters?

Revised in Section 2.3.5

5.The combinations of SVM-hyperparameters must be presented for each method.

 Revised in Table 2

6.Table 1
You showed the results of correlation analysis. Did you evaluate any reciprocal cooperation?

In this part, each feature is analyzed by correlation analysis, and features with high correlation coefficients are selected. Different feature combinations were used to evaluate classification accuracy later.

7. Discussion
A deeper discussion of the results in relation with similar works in literature in needed.

Revised in Line 376-383

Round 2

Reviewer 1 Report

I appreciate the correction of the manuscript and accept in present form.

Author Response

Revised in line 266&548. We have added relevant references to make the choice of RBF valid.

Reviewer 2 Report

Although the authors improved the manuscript well, they have to add some elements in the introduction section.
You said that the RBF kernel can realize non-linear mapping and has less numerical difficulties. But some references are missing.

It would be great to see your publication in this journal!

Author Response

Although the authors improved the manuscript well, they have to add some elements in the introduction section. 
You said that the RBF kernel can realize non-linear mapping and has less numerical difficulties. But some references are missing.

Revised in line 266&548,We have added relevant references to make the choice of RBF valid